# Evaluation and Multi-Objective Optimization of Lightweight Mortars Parameters at Elevated Temperature via Box–Behnken Optimization Approach

**DOI:** 10.3390/ma14237405

**Published:** 2021-12-02

**Authors:** Mehmet Kaya, Zeynel Baran Yıldırım, Fuat Köksal, Ahmet Beycioğlu, Izabela Kasprzyk

**Affiliations:** 1Department of Civil Engineering, Yozgat Bozok University, Yozgat 66100, Turkey; mehmet.kaya@yobu.edu.tr (M.K.); fuat.koksal@yobu.edu.tr (F.K.); 2Department of Civil Engineering, Dokuz Eylul University, Izmir 35390, Turkey; 3Department of Civil Engineering, Adana Alparslan Türkes Science and Technology University, Adana 01250, Turkey; abeycioglu@atu.edu.tr; 4Faculty of Civil and Environmental Engineering and Architecture, Bydgoszcz University of Science and Technology, 85-796 Bydgoszcz, Poland; izabela.kasprzyk@pbs.edu.pl

**Keywords:** lightweight mortar, silica fume, expanded vermiculite, response surface methodology, box-Behnken design

## Abstract

In this research, the mechanical properties of lightweight mortars containing different percentages of additional powder materials has been investigated using response surface methodology (RSM). Box–Behnken design, one of the RSM techniques, was used to study the effects of silica fume content (5, 10, and 15%), vermiculite/cement (V/C) ratio (4, 6, and 8), and temperature (300, 600, and 900 °C) on the ultrasonic pulse velocity (UPV), bending strength, and compressive strength of lightweight mortars. Design expert statistical software was accustomed to determining and evaluating the mix-design of materials in mortar mixtures and temperature effect on mortars. After preliminary experimental research of the relationships between independent and response variables, regression models were built. During the selection of the model parameters, F value, p-value, and R^2^ values of the statistical models were taken into account by using the backward elimination technique. The results showed a high correlation between the variables and responses. Multi-objective optimization results showed that the critical temperatures for different levels of silica fume (5–10–15%) were obtained as 371.6 °C, 306.3 °C, and 436 °C, respectively, when the V/C ratio kept constant as 4. According to the results obtained at high desirability levels, it is found that the UPS values varied in the range of 2480–2737 m/s, flexural strength of 3.13–3.81 MPa, and compressive strength of 9.9–11.5 MPa at these critical temperatures. As a result of this research, RSM is highly recommended to evaluate mechanical properties where concrete includes some additional powder materials and was exposed to high temperature.

## 1. Introduction

Due to the scarcity and insufficiency of natural resources, the need for energy has become one of the most important problems of today. In addition, it became difficult to find new energy sources. Therefore, energy efficiency has become an important issue. Energy is consumed in both heating and cooling of buildings. Thermal insulation in buildings is of great importance in terms of energy efficiency. Accordingly, research on new thermal insulation materials are still up-to-date. In addition to thermal insulation in buildings, fire resistance is a situation that should be considered in terms of building safety. Recently, research on the development of cement-based heat-insulating and fire-resistant lightweight composite materials have increasingly continued [1,2,3,4,5,6,7]. 

One of the most important methods known to produce lightweight concrete is to use lightweight aggregate. Another method is to produce concrete with polymer-based plastic granulated plastic aggregates [8]. Natural or industrial aggregates such as pumice, expanded clay, basalt powder and vermiculite, perlite, blast furnace slag, floor ash are used in concrete production [9,10,11]. Concretes produced with lightweight aggregates have positive properties such as heat [12,13,14] and sound insulation [15,16,17,18], reduced structure dead load [19,20,21], and high-temperature resistance [22,23,24] due to their low density. Lightweight mortars and concretes present some superior properties for structural applications such as fire resistance, thermal insulation, and sound insulation [15,16,25,26,27,28,29,30,31,32,33,34,35,36,37]. In addition, light mortars and concretes provide reductions in the cross-sections of the bearing elements and the earthquake loads acting on the structure by reducing the dead load of the structures [19,38,39].

One of the aggregates used in the production of high temperature resistant and lightweight concrete is vermiculite [13,22]. Vermiculite is a magnesium aluminosilicate clay mineral formed by the natural erosion of mica. Mineralogically, vermiculite, which represents a different group alone, is used as a general term covering all mica minarets (phlogopite, biotite, and hydrobiotite) which have industrial expansion properties. Vermiculite can also be described as aqueous magnesium, aluminum, iron silicate [40].

Vermiculite is a natural mineral and member of the Montmorillonite/Smectite Group. It is also placed in Clay and Mica Groups. Vermiculite is generally extremely hydrated biotite or phlogopite. These characteristic minerals later changed to vermiculite by weathering or hydrothermal processes. Vermiculite structures include water inside the interlayer cavities. The dilution properties are controlled via interlayered cations Mg^2+^ and small amounts of Ca^2+^, Na^+^, and K^+^. It influences the cation range and the level of hydration in the charge, intermediate and discharges layer arrays. The hydration status of vermiculite was determined by the quantity of water layers in the intermediate layer cavity. Water and cations among layers decide the thickness of the structural unit [41,42,43]. Vermiculites are divided into four groups as metamorphic vermiculite [44], macroscopic vermiculite, clay vermiculite [45], and autogenic vermiculite [46]. Vermiculite is crystallized in the monoclinic system and has a uniform slice. It can be green, yellowish coffee, or even black. Its hardness is between 1, 2 and 2.0 according to the Mohs scale and its specific weight is between 2.4 and 2.7. When vermiculite is suddenly subjected to heat-shock at high temperatures, it extends like an accordion. This characteristic expansion is thought to be due to the vapor pressure caused by the sudden evaporation of crystal water in the structure. The reason why thermal expansion has not yet been fully explained is that even samples containing the same total amount of water by weight can expand at different rates. Chemical bonding and the bonding of the water molecules between the leaves to the structure are other important parameters affecting the expansion event. As a result of the expansion, the bulk density of the material decreases by approximately 10 times from 0.8 g/ m^3^ to 0.08 g/cm^3^. The decrease in bulk density depends on the quality of the vermiculite and the performance of the furnace where the expansion is performed, and an approximate 30-fold expansion can be achieved as a result of heat treatment [47].

Silica fume, an industrial waste, is a very fine-grained powder obtained by the reduction of high-purity quartzite with coal and wood particles in electric arc furnaces used during the production of silicon metal or ferrosilicon alloys. In the upper parts of the furnaces at low temperatures, SiO gas is rapidly oxidized by contact with air and condenses as amorphous SiO2 to form almost all of the silica fume composition [48]. The unit weight of silica fume changes 250 and 300 kg/m³ [49]. Silica fume is composed of fine particles with a surface area of about 20,000 m²/kg [50]. It has been observed that compressive strength is increased [51,52], the interface is improved [53], and high-temperature resistance is increased in concretes produced with silica fume additive [21,52].

In recent years, experimental design methods have been extensively used in engineering studies. One of the experimental design methods is RSM which is used for the optimization of engineering problems and/or industrial processes. This methodology is a statistical method to investigate the best relationship between the dependent and independent parameters in experimental design and to determine their optimal use.

When the literature is examined, it is seen that this method has attracted the attention of the researchers especially in recent years and it has started to be used widely. Yıldırım et al. [54] studied to find optimal conditions for the effective waste coal additive on the effects of asphalt concrete utilizing a RSM strategy. Statistical analysis demonstrated that the model attained from the RSM study is appropriate for representing the best solution group model parameters. Miličević, Štirmer, and Bjegović [55], analyzed the effect of recycled aggregate on basic concrete properties (density, porosity e.g.,) by using Central Composite Design (CCD) and BBD. The comparison of values attained from prediction models with the experimental results showed that the BBD is feasible to find for basic properties of concrete including recycled aggregate and the number of experiments could be decreased. Rooholamini, Hassani, and Aliha [56], used CCD for choosing the best model of macro-synthetic fiber on the mechanical properties of roller-compacted concrete pavement (RCCP). The BBD methodology is accustomed to optimizing variables such as pH 3–7, the flow rate of 0.3–0.7 mL/min, and filter depth of 10–20 as well as seeing the effects of the determined parameters on column performance. The pH value has been shown to be the most important factor affecting the performance characteristics of fluorine removal with a fixed-bad column [57]. Adamu, Mohammed, and Liew [58] examined the effects of high-volume fly ash, crumb rubber, and nano-silica on roller-compacted concrete. Compressive, flexural, and splitting tensile strength were defined as response variables, and optimization of mixture proportions were carried out by using BBD design strategy. Analysis results demonstrated that nano-silica increases the performance and mechanical properties of high-volume fly ash samples of roller-compacted concrete with or without addition of crumb rubber. Asadzadeh and Khoshbayan [59] investigated the optimal conditions for foam concrete production including water, cement, and foam volume to obtain minimum density and maximum compressive strength along with the minimum cost. Total of 15 experimental samples were conducted, and the results were settled as response variables in statistical software for analyzing optimization. It was shown that the cost, compressive strength, and dry density response variables could be optimized simultaneously by BBD approach in foam concretes. Performance properties of the additives used in lightweight concretes have been examined by RSM methods and the usability of these techniques has been proven by many studies. Kockal and Ozturan investigated the optimization of the properties of lightweight fly ash aggregates in the production of high strength lightweight concrete using RSM. The effects of temperature, binder content and binder type independent parameters on specific gravity, water absorption, and crushing strength were evaluated and the process was optimized [60]. Using RSM techniques of lightweight mortars, the effect of curing temperatures, binder contents, and curing times on the compressive strength of geopolymer mortar [61], the effects of thermal permeability in different geometries on conduction and convection events [62], cement content, water ratio, and hydrogen peroxide ratios on pressure and the effects on bending strengths [63] were examined in detail in their studies.

The aim of the paper is to assess the effect of silica fume and expanded vermiculite on the behavior of mortars at elevated temperatures and establish a model depending on the test results by using a computer-based experimental program. It is aimed to see the effects of the changes at different levels of the independent variables on the dependent variables through the models created. In addition, multi-objective optimization cases will be evaluated at the desirability levels determined by simultaneous evaluation of dependent and independent variables. The novelty of this study is the evaluation of silica fume and expanded vermiculite mortars with RSM and the determination of critical temperatures in the conditions of multi-objective optimization.

## 2. Materials and Methods

CEM I 42.5R was used as cement obtained from Yibitaş cement plant, Yozgat, Turkey. Chemical and physical properties of cement are shown in Table 1.

The raw vermiculite procured from the Demircilik vermiculite deposit in Yıldızeli, Sivas, Turkey was used. Chemical properties of expanded vermiculite obtained by annealing raw vermiculite at 600 °C for a period of 10 s are given in Table 1. Physical properties of expanded vermiculite are also given in Table 1. Expanded vermiculite used in this research and its SEM image is given in Figure 1.

Besides the expanded vermiculite, in order to obtain higher compressive strength properties on mortars, silica fume additive was provided from Antalya Ferrochrome plants. Silica fume is an amorphous silica with a high specific surface area. The properties of silica fume used in this research are given in Table 1. As seen in Table 1, the total content of SiO_2_ + Al_2_O_3_ + Fe_2_O_3_ of silica fume is 89.16%. This is an important factor to increase the strength of mortars containing vermiculite. Ultrasound pulse velocities of specimens were determined according to EN 12504-4 standard [64], flexural and compressive tests were also made with respect to EN 1015-11 standard [65].

## 3. Experimental Design

### 3.1. Research Objective and Design Process

This study is aimed to investigate some mechanical properties of mortar affected by three inputs parameters. These inputs are silica fume content (5–10–15%), V/C ratio (4–6–8%), and temperature (300, 600, and 900 °C) which were selected as independent variables in experimental design. Within the design process, BBD approach, which is one of the response surface methodologies, was used as experimental design technique. Design Expert 10.0.2 experimental software was employed to generate design strategy, statistical analyses, and the optimization process. The flow chart of the design process shown in Figure 2 was followed step-by-step to select the appropriate design.

### 3.2. Theory of Experimental Design

RSM is the most appropriate and widely utilized statistical and numerical method used to analyze and develop models of one or more independent parameters that affect a process and the relationships between their responses. Moreover, RSM may be used in the multi-objective optimization model to define desirable targets based on either dependent or independent variables [66,67]. In RSM analysis, there are various design model types according to the suitability of the data used. Among these, the most commonly used methods are CCD and BBD. In RSM applications, there should be appropriate approaches to the interactions between response variables and independent variables. If the response variables are expressed in a linear model in terms of independent variables, the model equation given in Equation (1) is used [68,69,70].
*y* = *β*_0_ + *β*_1_*x*_1_ + *β*_2_*x*_2_ + … + *β_k_x_k_* + *ε*,(1)

However, if the curvature effect is significant or the experimental data does not fit to a first-order linear model, the linear model needs to be replaced by a second-order or higher order polynomial model seen in Equation (2).
(2)$$y=\beta_{0}+\overset{n}{\underset{i=1}{\sum }}\beta_{i}x_{i}+\overset{n}{\underset{i=1}{\sum }}\beta_{ii}x_{i}{}^{2}+\overset{n}{\underset{i\neq j=1}{\sum }}\beta_{ij}x_{i}x_{j}+\varepsilon$$

In this equation, *y* is the modeled response, *β* is the regression coefficient, *i* and *j* are the linear and quadratic coefficient, respectively. *x_i_* and *x_j_* are the coded values of independent variables and the term ε refers to experimental errors [58,66,67,70,71,72].

In RSM applications, desirability function is widely used to perform optimization of factors and response parameters. Desirability function, one of the popular methods used in simultaneously multi-objective optimization, was first introduced by Harrington [73] and further improved by Derringer and Suich [74]. Maximizing, minimizing, and target functions were utilized while optimizing the independent variables under the desired conditions [66,75,76]. The di values were calculated according to the desirability levels of the response variables. 1 represents the highest degree of desirability while 0 represents the lowest degree of desirability. Mathematical representation of the desirability function is given in Equation (3).
(3)$${D=[\overset{m}{\underset{r=1}{\prod }}d_{i}]}^{1/m}$$

### 3.3. Application of RSM by Using Box–Behnken Design Approach

The factors affecting the mix design are investigated by using BBD. The design is scheduled with the aid of utilizing BBD experimental technique by Design-Expert. Interactions between independent and dependent variables are analyzed by establishing mathematical models. Design matrix consists of 15 experimental series including 3 center points and 12 factorial points. The ranges and levels of the three factors –Silica Fume, V/C, and Temperature–are shown in Table 2.

In total, 15 experimental runs (3 replicates for the center point, 12 factorial points) in the randomized order were carried out for each response (Silica Fume, V/C, and Temperature). The design matrix generated using coded factors is given in Table 3.

In the design matrix planned using BBD, variables are defined as follows. The independent variables were Silica fume (*x*_1_), V/C ratio (*x*_2_), and temperature (*x*_3_); and the measured three responses were UPV (*y*_1_), bending strength (*y*_2_), and compressive strength (*y*_3_). The models set for each of the response variables are specified in the following equations.
(4)$$y_{1}=\beta_{1,0}+\beta_{1,1}x_{1}+\beta_{1,2}x_{2}+\beta_{1,3}x_{3}+\beta_{1,4}x_{1}x_{2}+\beta_{1,5}x_{1}x_{3+}\beta_{1,6}x_{2}x_{3}\;+\beta_{1,7}x_{1}{}^{2}+\beta_{1,8}x_{2}{}^{2}+\beta_{1,9}x_{3}{}^{2}$$
(5)$$y_{2}=\beta_{2,0}+\beta_{2,1}x_{1}+\beta_{2,2}x_{2}+\beta_{2,3}x_{3}+\beta_{2,4}x_{1}x_{2}+\beta_{2,5}x_{1}x_{3+}\beta_{2,6}x_{2}x_{3}\;+\beta_{2,7}x_{1}{}^{2}+\beta_{2,8}x_{2}{}^{2}+\beta_{2,9}x_{3}{}^{2}$$
(6)$$y_{3}=\beta_{3,0}+\beta_{3,1}x_{1}+\beta_{3,2}x_{2}+\beta_{3,3}x_{3}+\beta_{3,4}x_{1}x_{2}+\beta_{3,5}x_{1}x_{3+}\beta_{3,6}x_{2}x_{3}\;+\beta_{3,7}x_{1}{}^{2}+\beta_{3,8}x_{2}{}^{2}+\beta_{3,9}x_{3}{}^{2}$$
where $\beta_{1,0}$, $\beta_{2,0}$, $\beta_{3,0}$ are constant; $\beta_{1,1},\;\beta_{1,2},\beta_{1,3},\beta_{2,1},\beta_{2,2},\;\beta_{2,3},\;\beta_{3,1},\;\beta_{3,2},\beta_{3,3}$ are linear coefficients; $\beta_{1,4},\;\beta_{1,5}$, $\beta_{1,6}$ $\beta_{2,4},\;\beta_{2,5}$, $\beta_{2,6}$, $\beta_{3,4}$, $\beta_{3,5}$, $\beta_{3,6}$ are interactive coefficients; $\beta_{1,7},\;\beta_{1,8},\;\beta_{1,9},\;\beta_{2,6}$, $\beta_{2,7}$, $\beta_{2,8}$, $\beta_{2,9},$ $\beta_{3,7},\;\beta_{3,8},\;\beta_{3,9}$ are quadratic coefficients.

## 4. Laboratory Experiments according to Box–Behnken Design

In experimental design conducted by using BBD approach, it was recommended to prepare 15 different mortar mixes for laboratory experiments as seen in Table 3. The mechanical properties of these mortars found experimentally were needed as dependent variables in statistical analysis to find optimum design parameters. Expanded vermiculite aggregates were wetted with half of the water used in the mortar mixture one hour before mixing in the Hobart mixer. While mixing the mortars, vermiculite aggregates wetted after mixing silica fume and cement were added. Then the rest of the water required for the mixture was added.

After mixing all the materials for 3 min, fresh mortar samples were poured into molds of 40 × 40 × 160 mm size. Mortar specimens were kept at room temperature (20 °C) for 24 h. The specimens were then demolded and cured in 23 °C water for 27 days.

After the curing process was completed, the mortar samples were heated at 300, 600, and 900 °C. These temperatures are considered as critical temperatures for the cement paste to start the dehydration process. It is known that in case of fire, high-temperature exposure is only a few hours. For this reason, it is preferred to expose mortar samples to high temperature for 6 h. The temperature in the furnace was adjusted to increase by 5 °C per minute. During the heating of the samples in the furnace, the moisture in its contents are set free. After the heating process is completed, the mortar samples are allowed to cool slowly under 20 °C laboratory conditions. Then, bending and compressive strengths were determined according to TS EN 12390-5 [77] and TS EN 12390-3 [78] standards and UPV was determined according to ASTM C 597 [79].

## 5. Results and Discussions

Table 4 shows the BBD and the experimentally obtained responses (i.e., UPV, Bending Strength, and Compressive Strength). The value range from 2181 m/s to 2737 m/s, 0.90 MPa to 3.80 MPa, 3.50 MPa to 11.80 MPa for the UPV, bending strength and compressive strength, respectively.

In experimental design, variance analysis is used to evaluate whether independent parameters have a statistically significant effect on dependent parameters. In the statistical analysis, in addition to independent parameter effects (linear), two-factor interactions and quadratic form of independent parameters on dependent parameters may be observed, if it is appropriate to statistically significant levels [80,81,82]. In the analysis of variance, the contribution of statistically significant parameters to the model is evaluated by considering the predefined confidence level. In current study, the confidence interval was selected as 95%, which means that the p value was less than 0.05 (*p*-level < 0.05). In addition, the lack of fit is checked at the significance level of the *p*-value. If the lack of fit is statistically insignificant and *p* value of the lack of fit is greater than 0.05, then the model can be evaluated as significant. ANOVA results on response variables are given in Table 5.

The F-values of the models which found as −141.64 for UPV, 177.69 for Bending Strength, and 77.61 for Compressive Strength, demonstrate that the models were all significant, with only 0.09%, 0.01%, and 0.01% possibility, respectively. The significance of all models and terms was controlled using the 95% confidence interval (*p* < 0.05). For UPV, the model and terms B, AB, AC, C^2^, A^2^B, and AB^2^ were significant as their p values were <0.05, whereas A, C, BC, A^2^, and B^2^ were all insignificant. Bending strength model and its terms A, B, C, AC, BC, A^2^, C^2^, and A^2^ C were all significant as their Prob > F values were <0.05, whereas AB and B^2^ were all insignificant. For compressive strength, the model and terms A, B, C, AC, A^2^, B^2^, and C^2^ were all significant as their p values were <0.05, where AB and BC were all insignificant. The empirical models in terms of actual factors for UPV (*y*_1_), bending strength (*y*_2_), and compressive strength (*y*_3_) are presented in Equations (7)–(9).
(7)$$y_{1}=2391+8.0*A-173.75*B-3.12*C-45.25*AB-112.50*AC-10.25*BC\;+15.50*A^{2}+7.25*B^{2}+22.50*C^{2}-63.0*A^{2}B+33.25*AB^{2}$$
(8)$$y_{2}=2.60-0.0877*A-0.49*B-0.98*C+0.050*AB-0.17*AC+0.32*BC\;-0.15*A^{2}-0.58*C^{2}+0.20*A^{2}C$$
(9)$$y_{3}=7.90+0.74*A-2.37*B-1.74*C-0.20*AB-0.53*AC+0.25*BC-0.39\;*A^{2}+0.39*B^{2}-0.89*C^{2}$$

The final models’ equations were created by removing all insignificant terms for UPV (*y*_1_), bending strength (*y*_2_), and compressive strength (*y*_3_), respectively. These equations are given in Equations (10)–(12).
(10)$$y_{1}=2404.0-173.75*B-45.25*AB-112.50*AC+20.87*C^{2}-63.00*A^{2}B\;+41.25*AB^{2}$$
(11)$$y_{2}=2.60+0.087*A-0.49*B-0.98*C-0.17*AC+0.32*BC-0.15*A^{2}\;-0.58*C^{2}+0.20*A^{2}C$$
(12)$$y_{3}=7.90-0.74*A-2.37*B-1.74*C-0.53*AC-0.39*A^{2}+0.39*B^{2}-0.89\;*C^{2}$$

After selecting statistically significant parameters for each response variable, the regression model equations of these responses may be obtained. As mentioned in Equations (1)–(3), the interactions can be linear, two factor, and quadratic interactions, the most appropriate statistical model is found and accepted as the regression model of that response variable. The adjusted models for each of the response variables are specified in Equations (4)–(6). The negative and positive signs before a model term indicate the antagonistic or synergistic effects of independent variables on response variables.

The degree of correlation values was used to evaluate the adequacy and quality of the established models. Table 6 shows the coefficients of determination for responses investigated. In Table 6, all of the generated response variable models have significant R^2^ values that were greater than 0.85. Thus, nearly 99.38%, 99.68%, and 98.78% of the experimental data of the UPV, bending strength, and compressive strength models, respectively, can be correlated with the models. For the R^2^ values of the models to be in good agreement, the difference between the two should be <0.2. As can be seen differences between those values, it is seen that all response variables were in good agreement. In addition, the adequate precision (AP) values are given in Table 6. AP is a parameter that measures the signal to noise ratio, and it has to be greater than 4 to accept the desirability of responses. Considering the AP values, all models were in good agreement. The graphs indicating the relationship between predicted values from the established models and actual values are given in Figure 3a–c for UPV, bending strength, and compressive strength, respectively. As can be seen in Figure 3, the results obtained from the BBD model are very close to the experimental results.

A three-dimensional (3-D) surface graphs were used to present the relationship between two independent parameters and response parameters. Figure 4a shows 3-D response surface graphs of changes in UPV, the relationship between silica fume and V/C ratio when the temperature is taken as constant 300 °C. Figure 4b shows the relationship between silica fume and temperature, assuming the 3-D response surface graphs of the change in flexural strength, assuming the V/C ratio of 4. Figure 4c shows the 3-D response surface graphs of changes in compressive strength, the relationship between temperature and V/C ratio, when silica fume is taken as constant 10%.

For a better interpretation of the 3D response surface graphs, their 2D cases are also given. The color changes in these graphics represent the 3rd dimension. Blue values indicate low levels of the parameter, red values indicate high levels. Parabolic and linear effects can also be seen on these graphs.

When the response surface graphs are analyzed, the interaction between silica fume and V/C (A and B) (Figure 4a) is observed; at low levels of silica fume, the increasing V/C ratio has a parabolic decreasing effect on the UPV value, and at high levels of silica fume, the increase in the V/C ratio has a linear decreasing effect on the UPS value. In addition, increasing silica fume at all levels of the V/C ratio has a slightly increased parabolic effect on the UPS value.

In the interaction between silica fume and temperature (A and C) (Figure 4b), increasing the temperature at all levels of silica fume has a parabolic effect, which initially increases the bending strength value slightly and then decreases it. Moreover, increase in silica fume at all levels of temperature has a parabolic effect, which slightly decreases the bending strength value.

When the interaction between temperature and V/C ratio (B and C) is examined (Figure 4c), an increase in V/C ratio at all levels of temperature exhibits a linear decrease. Moreover, increasing in temperature at all levels of V/C ratio has a parabolic decrease in compressive strength value.

Simultaneously multi-objective optimization is the determination of the proportions of independent variables that can be used to obtain an optimized mortar mix based on the desired performance levels of response variables. In this study, optimization of response parameters was achieved by using desirability function in Design-Expert software version 10.0.2. In the simultaneous optimization criteria, critical temperatures were determined for all combinations of levels of 5–10–15% silica fume and 4–6–8 ratio of V/C. For this purpose, temperature and UPV variables were set to be in range, compressive and bending strength response variables were determined to be at maximum desirable levels as presented in Table 7.

Multi-objective optimization results are presented in Table 8. Desirability is the most important parameter to evaluate the success of optimization. As seen in the table, when the V/C ratio was kept constant as 4 and silica fume levels were chosen as 5%, 10%, and 15%, the critical temperatures were obtained as 371.6 °C, 306.3 °C, and 436 °C with the highest desirability percentages.

## 6. Conclusions

In this research, the effect of three independent parameters, namely, silica fume, V/C ratio, and temperature on the UPV, bending strength and compressive strength were investigated by using BBD approach. Following conclusions can be written:In the design approach, 15 experimental conditions were recommended by BBD. The recommended experimental conditions were applied at the laboratory and the results were found in the range of 2181–2737 m/s, 0.90–3.80 MPa, and 3.50–11.80 MPa for UPV, bending strength, and compressive strength, respectively.Statistical models were conducted on the results that were found in the recommended experiments. All models conducted on experimental results were found statistically significant according to p-values, R2 values, AP values, and lack of fit values.According to the relationship between dependent and independent variables observed in optimization, UPV value decreases when silica fume increases at high levels of temperature while the increase in silica fume at low levels of temperature exhibits a near-linear increase effect on UPV.Increasing temperature at all levels of silica fume has a parabolic effect, which initially increases the bending strength value slightly and then decreases it. Moreover, increasing in silica fume at all levels of temperature has a parabolic effect, which slightly decreases the bending strength value.Furthermore, a linear decrease was found in compressive strength when V/C increased. V/C ratio has the same effect on compressive strength at all levels of temperature. A similar decrease in compressive strength was found for an increase in temperature at all levels of V/C ratio.An optimization was made to find the maximized mechanical performances of mortar by using statistical models. When the V/C ratio was kept constant as 4 and silica fume levels were chosen as 5%, 10%, and 15%, the critical temperatures were obtained as 371.6 °C, 306.3 °C, and 436 °C with the highest desirability percentages.

As a result of this study, it is seen that the experimental design methods can be very useful for laboratory studies to decrease labor efforts, materials consumption, cost and time period.

## Figures and Tables

**Figure 1 materials-14-07405-f001:**
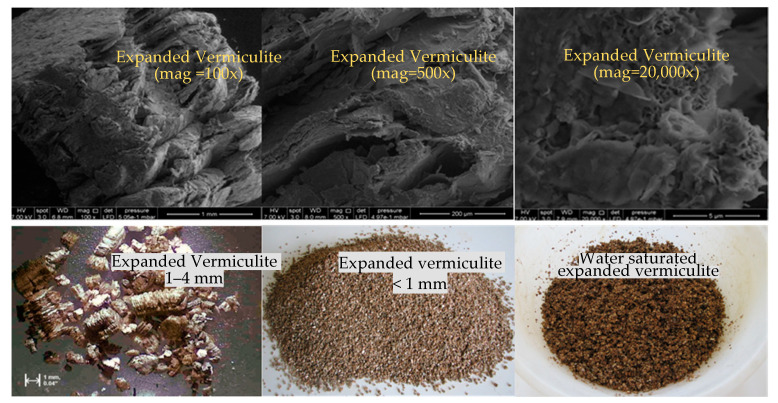
SEM images of expanded vermiculite and the form before use in the mixture.

**Figure 2 materials-14-07405-f002:**
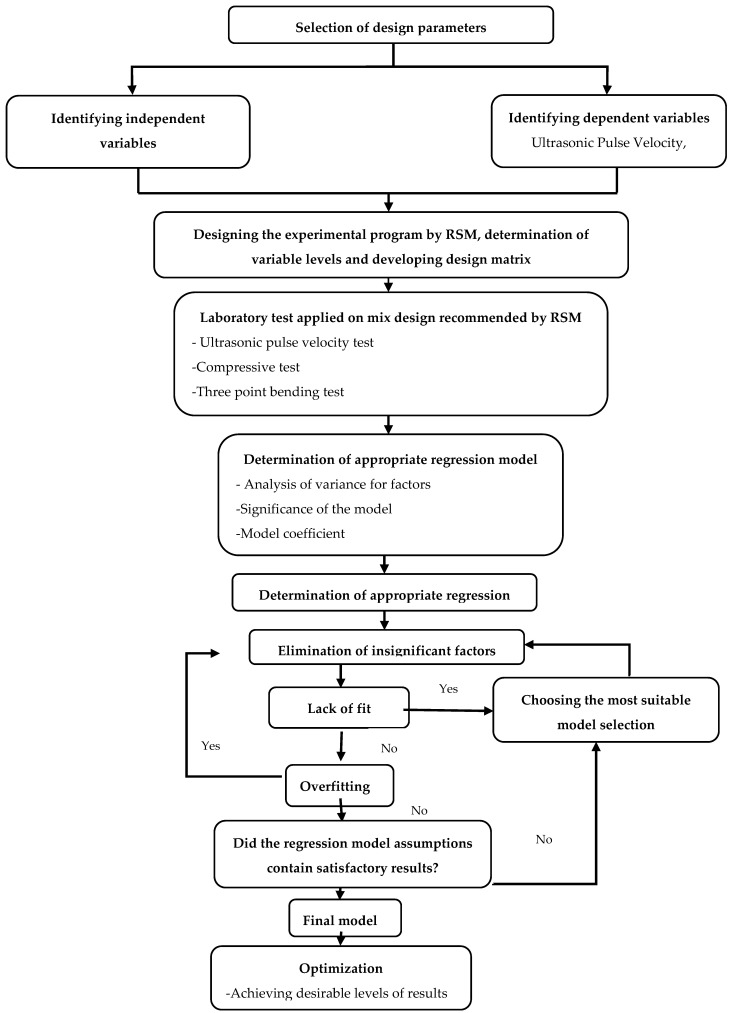
Research process.

**Figure 3 materials-14-07405-f003:**
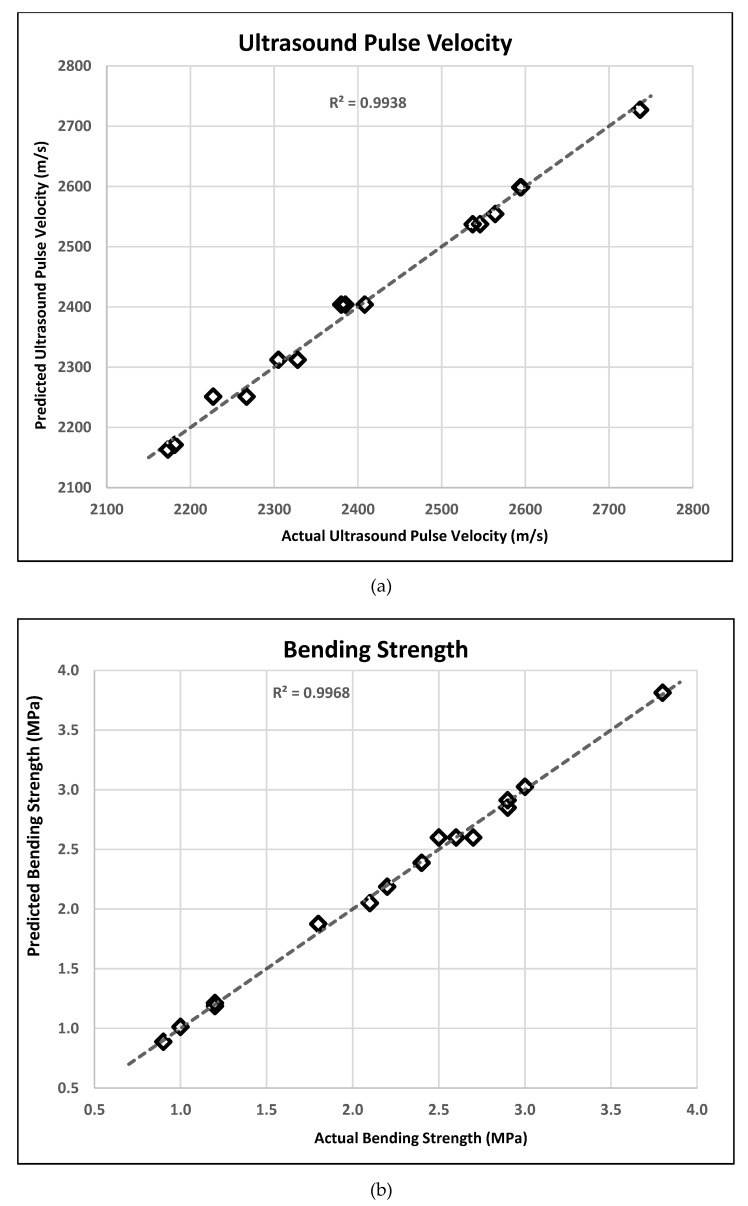
Relationship between experimental and predicted values of response variable.

**Figure 4 materials-14-07405-f004:**
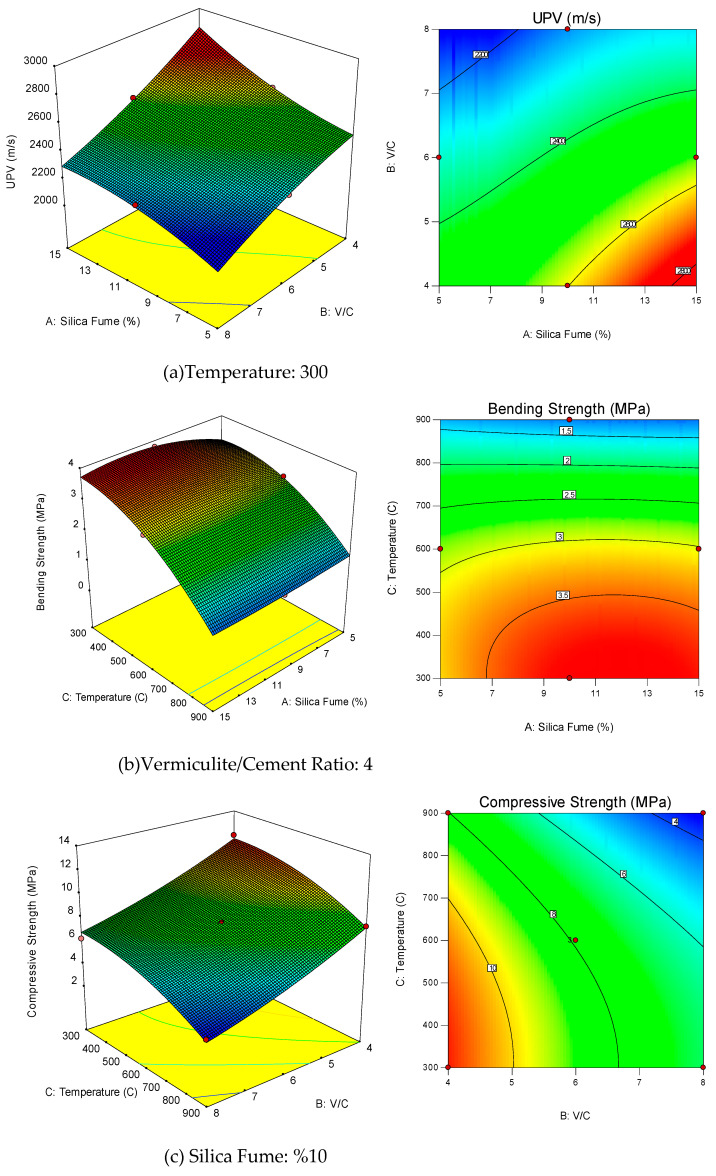
Influence of independent variables and their interactions on response variables.

**Table 1 materials-14-07405-t001:** Properties and characteristics of CEM I 42.5 Portland cement, expanded vermiculite, and silica fume.

**Chemical Composition of CEM I 42.5 R Portland Cement (%)**	
MgO	2.75
Si_2_O_2_	19.12
AI_2_O_3_	5.63
Fe_2_O_3_	2.39
Na_2_O	-
CaO	63.17
SO_3_	2.74
K_2_O	1
Insoluble materials	2.33
Loss on ignition	0.49
**Physical Composition of CEM I 42.5 R Portland Cement**	
Specific gravity (g/cm^3^)	3.09
Blaine specific surface area (cm^2^/g)	3114
Initial setting time (min)	150
Final setting time (min)	215
**Chemical properties and some characteristics of expanded vermiculite**	
Color	Gold
Combustibility	Non-combustible
Shape	According to shape granule
Cation exchange capacity	50–150 meg/100 g
Permeability	95%
Water holding capacity	240% (by weight)
	28% (by volume)
pH	8.1
Thermal conductivity	0.066–0.063 W/m K
Bulk density	140 kg/m^3^
Specific gravity	0.22
Specific heat	0.20–0.26 kcal/kg °C
Sintering temperature	1150–1250 °C
SiO_2_	36.90%
AI_2_O_3_	17.70%
CaO	3.50%
TiO_2_	2.20%
MgO	16.40%
K_2_O	2.60%
Na_2_O	0.20%
Fe_2_O_3_	11.20%
Loss on ignition	9.20%
**Physical and chemical properties of silica fume**	
MgO	1.47
AI_2_O_3_	1.42
CaO	0.8
SO_3_	1.34
SiO_2_ + AI_2_O_3_ + Fe_2_O_3_	89.16
SiO_2_	85.35
Fe_2_O_3_	2.39
Loss on ignition	3.4
Moisture (%)	0.19
Bulk density	0.55–0.65 kg/dm^3^
Retaining on 45-micron sieve (%)	0.58
Specific gravity	2.23
Surface area (cm^2^/g)	8900

**Table 2 materials-14-07405-t002:** Independent variables and their levels.

		Levels of Code		
Code	Factor	−1	0	+1
A	Silica Fume, %	5	10	15
B	V/C	4	6	8
C	Temperature, °C	300	600	900

**Table 3 materials-14-07405-t003:** Box–Behnken Design for three variables.

Experiment/Mix No.	A	B	C
1	−1	1	0
2	1	1	0
3	0	1	1
4	−1	0	1
5	0	0	0
6	0	1	−1
7	1	−1	0
8	−1	0	−1
9	0	−1	1
10	−1	−1	0
11	0	0	0
12	1	0	−1
13	1	0	1
14	0	−1	−1
15	0	0	0

**Table 4 materials-14-07405-t004:** Design matrix and responses obtained.

Run No	Silica Fume (%)	V/C	Temperature °C	UPV (m/s)	Bending Strength (MPa)	Compressive Strength (MPa)
	*x* _1_	*x* _2_	*x* _3_	*y* _1_	*y* _2_	*y* _3_
**1**	5	8	600	2181	1.8	5.3
**2**	15	8	600	2173	2.1	6.1
**3**	10	8	900	2227	0.9	3.5
**4**	5	6	900	2537	1.2	4.4
**5**	10	6	600	2380	2.5	7.9
**6**	10	8	300	2267	2.2	6.2
**7**	15	4	600	2737	3.0	10.9
**8**	5	6	300	2305	2.4	7.1
**9**	10	4	900	2595	1.2	8.1
**10**	5	4	600	2564	2.9	9.3
**11**	10	6	600	2385	2.6	7.8
**12**	15	6	300	2546	2.9	9.9
**13**	15	6	900	2328	1.0	5.1
**14**	10	4	300	2594	3.8	11.8
**15**	10	6	600	2408	2.7	8.0

**Table 5 materials-14-07405-t005:** ANOVA for developed response models.

Source	Sum of Squares	df	Mean Square	F Value	Prob > F (*p*-Value)
** *Ultrasonic Pulse Velocity (m/s)* **					
Model	413,984.6	11	37,635.0	141.64	**0.0009**
A-Silica Fume	256.0	1	256.0	0.96	0.3987
B-V/C	120,756.3	1	120,756.3	454.47	0.0002
C-Temperature	78.1	1	78.1	0.29	0.6253
AB	8190.2	1	8190.2	30.82	0.0115
AC	50,625.0	1	50,625.0	190.53	0.0008
BC	420.3	1	420.3	1.58	0.2975
A^2^	887.1	1	887.1	3.34	0.1651
B^2^	194.1	1	194.1	0.73	0.4556
C^2^	1869.2	1	1869.2	7.03	0.0768
A^2^B	7938.0	1	7938.0	29.87	0.0120
AB^2^	2211.1	1	2211.1	8.32	0.0633
Residual	797.1250	3	265.7		
Lack of Fit	351.1250	1	351.1	1.57	**0.3363**
** *Bending Strength (MPa)* **					
Model	9.9948	10	0.9995	177.69	**0.0001**
A-Silica Fume	0.0612	1	0.0612	10.89	0.0299
B-V/C	1.9013	1	1.9013	338.00	0.0001
C-Temperature	3.8025	1	3.8025	676.00	0.0000
AB	0.0100	1	0.0100	1.78	0.2533
AC	0.1225	1	0.1225	21.78	0.0095
BC	0.4225	1	0.4225	75.11	0.0010
A^2^	0.0831	1	0.0831	14.77	0.0184
B^2^	0.0000	1	0.0000	0.00	1.0000
C^2^	1.2208	1	1.2208	217.03	0.0001
A^2^C	0.0800	1	0.0800	14.22	0.0196
Residual	0.0225	4	0.0056		
Lack of Fit	0.0025	2	0.0013	0.13	**0.8889**
** *Compressive Strength (MPa)* **					
Model	79.2818	9	8.8091	77.61	**0.0001**
A-Silica Fume	4.3513	1	4.3513	38.34	0.0016
B-V/C	45.1250	1	45.1250	397.58	0.0000
C-Temperature	24.1513	1	24.1513	212.79	0.0000
AB	0.1600	1	0.1600	1.41	0.2884
AC	1.1025	1	1.1025	9.71	0.0264
BC	0.2500	1	0.2500	2.20	0.1979
A^2^	0.5544	1	0.5544	4.88	0.0781
B^2^	0.5544	1	0.5544	4.88	0.0781
C^2^	2.9083	1	2.9083	25.62	0.0039
Residual	0.5675	5	0.1135		
Lack of Fit	0.5475	3	0.1825	18.25	**0.0524**

where A: Silica fume, B: Vermiculite/Cement ratio, C: Temperature, A^2^, B^2^ and C^2^: second order effect, A*B, A*C and B*C two factor interaction effects, A^2^B, AB^2^ and A^2^C qubic effects df: Degree of freedom, F-values: Fisher-statistical test value, *p*-values: Probability values.

**Table 6 materials-14-07405-t006:** Coefficient of determinations for response variables.

Response	R^2^	Adj R^2^	Pred R^2^	A.P.
Ultrasonic Pulse Velocity (m/s)	0.9938	0.9891	0.9094	46.020
Bending strength (Mpa)	0.9968	0.9924	0.9893	51.304
Compressive strength (Mpa)	0.9878	0.9755	0.9287	30.139

where R^2^: degree of correlation, Adj R^2^: adjusted the degree of correlation, Pred R^2^: predicted degree of correlation, A.P: adequate precision.

**Table 7 materials-14-07405-t007:** Multi-objective optimization criteria.

Variables and Responses	Symbol	Goal	Lower Limit	Upper Limit
Silica Fume (%)	A	(%5–10–15)	5	15
V/C	B	(4–6–8)	4	8
Temperature °C	C	In range	300	900
Ultrasonic Pulse Velocity (m/s)	*y_1_*	In range	2173	2737
Bending Strength (MPa)	*y_2_*	Maximize	0.9	3.8
Compressive Strength (MPa)	*y_3_*	Maximize	3.5	11.8

**Table 8 materials-14-07405-t008:** Optimization with desirability values.

Solution Number	Silica Fume (%)	V/C	Temperature °C	UPV (m/s)	Bending Strength (MPa)	Compressive Strength (MPa)	Desirability (%)
1	5.00	4.00	371.6	2480.7	3.22	9.9	78.9
2	10.00	4.00	306.3	2597.8	3.81	11.5	98.3
3	15.00	4.00	436.605	2737	3.13	11.2	84.4
4	5.00	6.00	423.9	2345.2	2.52	7.2	49.7
5	10.00	6.00	330.2	2420.8	3.01	8.7	67.8
6	15.00	6.00	306.8	2533.9	2.91	9.6	71.6
7	5.00	8.00	604.7	2173	1.87	4.7	22.6
8	10.00	8.00	380.89	2241.4	2.28	6.7	42.9
9	15.00	8.00	364.1	2264.6	2.19	7.5	46.2

## Data Availability

Data are summarized in Table 4. They can be requested by contacting the corresponding author.

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
