# Peer review of "Evaluation and Multi-Objective Optimization of Lightweight Mortars Parameters at Elevated Temperature via Box–Behnken Optimization Approach"

_materials, 2021, doi:10.3390/ma14237405_

Round 1

Reviewer 1 Report

As attached

Author Response

Dear Editor;

We sincerely thank the reviewers for constructive criticisms and valuable comments, which were of great help in revising the manuscript. Accordingly, the revised manuscript has been systematically improved with new information and additional interpretations. Our responses to the reviewer’s comments are given below.

RESPONSES TO THE RECOMMENDATIONS

(REVIEWER 1)

Q1: The title so confusing and not clear, authors should provide new suitable title for this manuscript.

Response to Q1: The title has been changed as follows;

“Evaluation and multi-objective optimization of lightweight mortars parameters at elevated temperature via Box-Behnken optimization approach”

Q2: Abstract: This study offers with the utilization of Response Surface Methodology (RSM) to layout, advance statistical models, and determine the optimization for the mixtures the use of the variables Silica Fume content (5-10-15%), Please revise this sentence.

Response to Q2:  

“This study offers with the utilization of Response Surface Methodology (RSM)  to layout, advance statistical models, and determine the optimization for the mixtures the use of the variables Silica Fume content (5-10-15%), Vermiculite/Cement (V/C) ratio (4-6-8%) and Temperature (300-600-900°C) in order to achieve Ultrasonic Pulse Velocity (UPV) in range and maximum Bending Strength, and Compressive Strength. Within the design process, Box-Behnken Design (BBD) approach, which is one of the response surface methodologies, was employed as an experimental design technique.”

The above sentences in the summary have been updated as follows.

“In this research, the mechanical properties of lightweight mortars containing different percentages of additional powder materials has been investigated using response surface methodology (RSM). Box-Behnken Design, one of the RSM techniques, was used to study the effects of silica fume content (5, 10 and 15%), vermiculite/cement (V/C) ratio (4, 6 and 8), and temperature (300, 600 and 900°C) on the ultrasonic pulse velocity (UPV), bending strength, and compressive strength of lightweight mortars.”

Q3: (300-600-900°C) revise to (300, 600 and 900).

Response to Q3:  The expressions in the text have been changed as suggested by the referee.

Line 18

Line 176

Line 264

Q4: The last part in abstract should revise by authors and show the critical findings for this study.

Response to Q4:  In the last part of the summary, some of the critical findings of the study are added as follows.

“Multi-objective optimization results showed that the critical temperatures for different levels of silica fume (5-10-15%) were obtained as 371,6°C, 306,3°C, and 436°C, respectively, when the V/C ratio kept constant as 4. According to the results obtained at high desirability levels, it is found that the UPS values varied in the range of 2480-2737 m/s, flexural strength of 3.13-3.81 MPa and compressive strength of 9.9-11.5 MPa at these critical temperatures.”

Q5: First paragraph in introduction section not well present, poor in writing and information, please revise this paragraph.

Response to Q5:

First paragraph in introduction section was revised as follows:

“Due to the scarcity and insufficiency of natural resources, the need for energy has be-come one of the most important problems of today. In addition, it became difficult to find new energy sources. Therefore, energy efficiency has become an important issue. Energy is consumed in both heating and cooling of buildings. Thermal insulation in buildings is of great importance in terms of energy efficiency. Accordingly, researches on new thermal insulation materials are still up-to-date. In addition to thermal insulation in buildings, fire resistance is a situation that should be considered in terms of building safety. Recently, researches on the development of cement-based heat-insulating and fire-resistant light-weight composite materials have increasingly continued. [1-7]”

Q6: Authors should provide more literature about sustainable applications of lightweight mortar and concrete.

Response to Q6: More extensive literature on lightweight mortars and concretes has been added.

Following paragraph was added to introduction section:

“Lightweight mortars and concretes present some superior properties for structural applications such as fire resistance, thermal insulation and sound insulation [25-39]. In addi-tion, light mortars and concretes provide reductions in the cross-sections of the bearing elements and the earthquake loads acting on the structure by reducing the dead load of the structures [40-42].”

Q7: Please highlight your research novelty, limitation and aims.

Response to Q7: The explanations about the aims, limitations and novelty of the research are expanded in a more comprehensive way in the last paragraph of the introduction. In the last part of the introduction, the paragraph is as follows;

“The aim of the paper is to assess the effect of silica fume and expanded vermiculite on the behavior of mortars at elevated temperatures and establish a model depending on test results by using a computer-based experimental program. It is aimed to see the effects of the changes at different levels of the independent variables on the dependent variables through the models created. In addition, multi-objective optimization cases will be evaluated at the desirability levels determined by simultaneous evaluation of dependent and independent variables. The novelty of this study is the evaluation of silica fume and expanded vermiculite mortars with RSM and the determination of critical temperatures in the conditions of multi-objective optimization.”

Q8: Please combine the chemical composition of OPC, silica fume and vermiculite in one table.

Response to Q8: Table 1, Table 2 and Table 3 were combined. The final state of the table is as follows;

                Table 1.Properties and characteristics of CEM I 42.5 Portland cement, expanded vermiculite and silica fume

Chemical Composition of CEM I 42.5 R Portland cement (%)

MgO

2,75

Si2O2

19,12

AI2O3

5,63

Fe2O3

2,39

Na2O

-

CaO

63,17

SO3

2,74

K2O

1

Insoluble materials

2,33

Loss on ignition

0,49

Physical Composition of CEM I 42.5 R Portland cement

Specific gravity (g/cm3)

3,09

Blaine specific surface area (cm2/g)

3114

Initial setting time (min)

150

Final setting time (min)

215

Chemical properties and some characteristics of expanded vermiculite

Color

Gold

Combustibility

Non-combustible

Shape

According shape granule

Cation exchange capacity

50-150 meg/100 g

Permeability

95%

Water holding capacity

240% (by weight)

28% (by volume)

pH

8,1

Thermal conductivity

0,066-0,063 W/m K

Bulk density

140 kg/m3

Specific gravity

0,22

Specific heat

0,20-0,26 kcal/kg°C

Sintering temperature

1150-1250°C

SiO2

36,90%

AI2O3

17,70%

CaO

3,50%

TiO2

2,20%

MgO

16,40%

K2O

2,60%

Na2O

0,20%

Fe2O3

11,20%

Loss on ignition

9,20%

Physical and chemical properties of silica fume

MgO

1,47

AI2O3

1,42

CaO

0,8

SO3

1,34

SiO2 + AI2O3 + Fe2O3

89,16

SiO2

85,35

Fe2O3

2,39

Loss on ignition

3,4

Moisture (%)

0,19

Bulk density

0,55-0,65 kg/dm3

Retaining on 45-micron sieve (%)

0,58

Specific gravity

2,23

Surface area (cm2/g)

8900

Q9: Figure 1. SEM images: please provide more details about this figure (a, b and c?)

Response to Q9:

More details about Figure 1 were provided and explanation of each figure was showed on figure. The figure is given below.

Expanded Vermiculite 1- 4 mm

Expanded vermiculite < 1 mm

Water saturated

expanded vermiculite

Expanded Vermiculite

(mag=20000x)

Expanded Vermiculite

(mag=500x)

Expanded Vermiculite

(mag =100x)

Q10: Fig. 2: Please provide high quality image.

Response to Q10:  Figure 2 has been added to the article in higher resolution.

Q11: Fig. 3, not important, please remove it.

Response to Q11:

Fig. 3. was removed from manuscript and its statement in the text was also removed.

Q12: More details about test methods should provide by authors.

Response to Q12:

Following statement were added to text:

“Ultrasound pulse velocities of specimens were determined according to EN 12504-4 standard [67], flexural and compressive tests were also made with respect to EN 1015-11 standard [68”

Q13: Very poor discussion provided by authors. The results section should re-discuss by authors with more information and details.

Response to Q13: In the results and discussions section, the following topics are examined in detail.

- Samples were produced according to the conditions determined in the experimental design matrix. Experimental results were used in statistical models.

- All the details of constructing a significant statistical model were discussed. In addition, the models created for all independent parameters and all the results of their parameters are given in the ANOVA table.

-Equations obtained from the statistical model and their coefficients were discussed in detail.

-R2 values ​​were discussed and predicted-actual graphs were drawn.

-By adding 2 and 3D surface response graphics, evaluations for the levels of independent and dependent variables were made and discussed.

-Optimization conditions were discussed and the results were evaluated in detail.

When the articles about RSM techniques are examined in the references section, we think that the results and discussion section of our article are much stronger. Important issues in terms of performance characteristics were evaluated and discussed. According to other suggestions made by the reviewers, the results and discussion section were further strengthened by giving details.

Q14: Fig. 4: Provide high quality image.

Response to Q14: Figure 2 has been added to the article in higher resolution.

Q15: Fig. 5: not well discussed and authors should re-discuss this figure.

Response to Q15: Logic errors were corrected in the paragraphs in which Figure 5 was interpreted. The graphics are discussed in more detail and comprehensively in the text. For all variations of the independent variables, their effects on the dependent variables are indicated in detail. In addition to 3-dimensional graphics, it is stated how 2-dimensional graphics should be interpreted. The paragraphs in which Figure 5 is discussed are arranged as follows.

“A three-dimensional (3-D) surface graphs were used to present the relationship between two independent parameters and response parameters. Figure 5a shows 3-D response surface graphs of changes in UPV, the relationship between silica fume and V/C ratio when the temperature is taken as constant 300°C. Figure 5b shows the relationship between silica fume and temperature, assuming the 3-D response surface graphs of the change in flexural strength, assuming the V/C ratio of 4. Figure 5c shows the 3-D response surface graphs of changes in compressive strength, the relationship between temperature and V/C ratio, when silica fume is taken as constant 10%.

For a better interpretation of the 3D response surface graphs, their 2D cases are also given. The color changes in these graphics represent the 3rd dimension. Blue values in-dicate low levels of the parameter, red values indicate high levels. Parabolic and linear effects can also be seen on these graphs.

When the response surface graphs are analyzed, it is observed that that the interac-tion between silica fume and V/C (A and B) (Figure 5a); At low levels of silica fume, the increasing V/C ratio has a parabolic decreasing effect on the UPV value, and at high levels of silica fume, the increase in the V/C ratio has a linear decreasing effect on the UPS value. In addition, increasing silica fume at all levels of the V/C ratio has a slightly increased parabolic effect on the UPS value.

In the interaction between silica fume and temperature (A and C) (Fig. 5b); Increasing temperature at all levels of silica fume has a parabolic effect, which initially increases the bending strength value slightly and then decreases it. Also, increasing in silica fume at all levels of temperature has a parabolic effect, which slightly decrease the bending strength value.

When the interaction between temperature and V/C ratio (B and C) is examined (Fig. 5c); while increasing in V/C ratio at all levels of temperature exhibits a linear decrease. Moreover, increasing in temperature at all levels of V/C ratio has a parabolic decrease in compressive strength value.”

Reviewer 2 Report

Novelty of this study should be clearly described in the Introduction section. 

From experience, large data is needed to obtain an accurate statistical model, how come the study only considered 15 samples?

Improve the discussion with relevant related research works.

References should be done according to the journal format.

Author Response

Dear Editor;

We sincerely thank the reviewers for constructive criticisms and valuable comments, which were of great help in revising the manuscript. Accordingly, the revised manuscript has been systematically improved with new information and additional interpretations. Our responses to the reviewer’s comments are given below.

RESPONSES TO THE RECOMMENDATIONS

(REVIEWER 2)

Q1: Novelty of this study should be clearly described in the Introduction section.

Response to Q1:  The explanations about novelty of the research are expanded in a more comprehensive way in the last paragraph of the introduction section. In the last part of the introduction, the paragraph is as follows;

“The aim of the paper is to assess the effect of silica fume and expanded vermiculite on the behavior of mortars at elevated temperatures and establish a model depending on test results by using a computer-based experimental program. It is aimed to see the effects of the changes at different levels of the independent variables on the dependent variables through the models created. In addition, multi-objective optimization cases will be evaluated at the desirability levels determined by simultaneous evaluation of dependent and independent variables. The novelty of this study is the evaluation of silica fume and expanded vermiculite mortars with RSM and the determination of critical temperatures in the conditions of multi-objective optimization.”

Q2: From experience, large data is needed to obtain an accurate statistical model, how come the study only considered 15 samples?

Response to Q2:  RSM methods are used in experimental studies used in many engineering fields. One of the biggest advantages of this method is that it provides savings in terms of both time and cost by reducing the number of experiments required under normal conditions. Considering CCD and BBH, which are among the most known RSM techniques, a large percentage of the experimental studies in the literature consist of an average of 15-25 experimental sets. It does not need all the data in experimental combinations to find the effects of independent variables on dependent variables. Models are created using fewer data sets and these models are supported statistically. These effects are seen in the ANOVA analysis. There are many studies that do not require large data sets to create statistical models. When we look at the studies examined in the introduction section, this situation can be seen.

Q3: Improve the discussion with relevant related research works.

Response to Q3:  Research articles using RSM method in the fields of contruction materials and concrete technologies were given in the introduction and experimental design sections of our study. When the citations from the literature are examined, it will be seen that most of the studies in which RSM is used show similarities to the graphics and results mentioned in the discussion section. Since the model outputs already contain many analyzes, evaluations and comments are included in the discussion section.

Q4: References should be done according to the journal format.

Response to Q4:  References in the article were arranged according to the journal format.

Reviewer 3 Report

Dear Authors.

Please do the following corrections.

Abstract: It is obvious that RSM will works fine when there is a relation between input factors and their responses.  But you have to add how it gives you profit on your research outcome.  Lines 23-26: These are just overall common statement to say RSM works fine for you. It will be good for readers if you add your results findings like which parameters positively or negatively impact the response (output). You can make this statement based on your regression equation and the +/- signs stands in front of the input parameters. You can make clear results findings like which affects the most main terms or interaction terms. Please interpret the results and discussion part and make few lines in the abstract to make it more friendly to the readers.

Introduction: Lines: 123--125: Try to add more information like what is the difference in your research compared to other published information's. Add more information rather than just mentioning outline of your research. refer to other published articles.

Line 173: Y variables should be in italic form. Please update all the variables into italic form.

Line 177:  Xi and Xj: should be italic and subscript.

Line 201-24: change variables to italic form and update with proper subscript.

remove commas put dots For ex. 3,50 MPa to 3.5o MPa. It will be more reasonable. Remove in Tables as well as in text.

results are discussed well and appropriate. 

Conclusion looks good.

The paper can be accepted after the minor corrections. 

Author Response

Dear Editor;

We sincerely thank the reviewers for constructive criticisms and valuable comments, which were of great help in revising the manuscript. Accordingly, the revised manuscript has been systematically improved with new information and additional interpretations. Our responses to the reviewer’s comments are given below.

RESPONSES TO THE RECOMMENDATIONS

(REVIEWER 3)

Q1: Abstract: It is obvious that RSM will works fine when there is a relation between input factors and their responses.  But you have to add how it gives you profit on your research outcome.  Lines 23-26: These are just overall common statement to say RSM works fine for you. It will be good for readers if you add your results findings like which parameters positively or negatively impact the response (output). You can make this statement based on your regression equation and the +/- signs stands in front of the input parameters. You can make clear results findings like which affects the most main terms or interaction terms. Please interpret the results and discussion part and make few lines in the abstract to make it more friendly to the readers.

Response to Q1:  The following underlined statement has been added to the summary section. Since the selection of model parameters is given in detail in the ANOVA analysis, numerical details are not included in the summary. All of the analyzes mentioned in the underlined sentence are given in the results and discussion section and their graphics are evaluated. The model coefficients that affect the response parameters are also clearly stated in the results and discussions section. The detailed selection and explanations of the model parameters affecting each of the response variables were also evaluated in the ANOVA analysis.

“…Design Expert statistical software was accustomed to determine and evaluate the mix-design of materials in mortar mixtures and temperature effect on mortars. After preliminary experimental research of the relationships between independent and response variables, regression models were built. During the selection of the model parameters, F value, p-value, and R2 values of the statistical models were taken into account by using the Backward elimination technique. The results showed a high correlation between the variables and responses…”

Q2: Introduction: Lines: 123--125: Try to add more information like what is the difference in your research compared to other published information's. Add more information rather than just mentioning outline of your research. refer to other published articles.

Response to Q2: Studies carried out using RSM techniques in lightweight concretes are added to the Introduction section. The parameters examined in these studies and the optimization details are given. The following paragraph has been added.

“Performance properties of the additives used in lightweight concretes have been examined by RSM methods and the usability of these techniques has been proven by many studies. Kockal and Ozturan investigated the optimization of the properties of lightweight fly ash aggregates in the production of high strength lightweight concrete using RSM. The effects of temperature, binder content and binder type independent parameters on specific gravi-ty, water absorption and crushing strength were evaluated and the process was optimized [63]. Using RSM techniques of lightweight mortars; The effect of curing temperatures, binder contents and curing times on the compressive strength of geopolymer mortar [64], the effects of thermal permeability in different geometries on conduction and convection events [65], cement content, water ratio, and hydrogen peroxide ratios on pressure and the effects on bending strengths [66] were examined in detail in their studies.”

 The explanations about the aims, limitations and novelty of the research are expanded in a more comprehensive way in the last paragraph of the introduction. In the last part of the introduction, the paragraph is as follows;

“The aim of the paper is to assess the effect of silica fume and expanded vermiculite on the behavior of mortars at elevated temperatures and establish a model depending on test results by using a computer-based experimental program. It is aimed to see the effects of the changes at different levels of the independent variables on the dependent variables through the models created. In addition, multi-objective optimization cases will be evaluated at the desirability levels determined by simultaneous evaluation of dependent and independent variables. The novelty of this study is the evaluation of silica fume and expanded vermiculite mortars with RSM and the determination of critical temperatures in the conditions of multi-objective optimization.”

Q3: Line 173: Y variables should be in italic form. Please update all the variables into italic form.

Response to Q3:  All variables are shown in italics.

Q4: Line 177:  Xi and Xj: should be italic and subscript

Response to Q4:  Xi and Xj shown in italics and subscripts.

Q5: Line 201-24: change variables to italic form and update with proper subscript.

Response to Q5:  All the edits have been made.

Q6: remove commas put dots For ex. 3,50 MPa to 3.5o MPa. It will be more reasonable. Remove in Tables as well as in text.

Response to Q6: The commas on the numeric data were arranged.

Round 2

Reviewer 1 Report

  1. Authors not follow the previous comments to improve the manuscript content, please clarify.
  2. Provided title not suitable for this articles.

Reviewer 2 Report

Authors have improved the article